# Zero-Cost Virtual RNA: Approximating Immunotherapy Signatures via Cross-Modal WSI Retrieval

**Sigrid Vila-Bagaria**[1]  ID                                SIGRID.VILA@UPC.EDU
**Mar Teixidó**[2]  ID                                      MTEIXIDO@IRBLLEIDA.CAT
**Miquel Piñol**[2]  ID                               MPINOLR.LLEIDA.ICS@GENCAT.CAT
**Felip Vilardell**[2]  ID                          FVILARDELL.LLEIDA.ICS@GENCAT.CAT
**Robert Montal**[2]  ID                             RMONTAL.LLEIDA.ICS@GENCAT.CAT
**Verónica Vilaplana**[1]  ID                             VERONICA.VILAPLANA@UPC.EDU

[1] *Image Processing Group, Universitat Politècnica de Catalunya, Spain*

[2] *Research group of Cancer Biomarkers & Oncological Pathology, IRB Lleida, Spain*

## Abstract

Identifying the "Inflamed" immunophenotype in Gastric Adenocarcinoma predicts immunotherapy response but requires an expensive 10-gene RNA signature. While deep learning on standard H&E slides offers a scalable alternative, conventional binary classifiers oversimplify continuous RNA data and introduce label noise. To resolve this, we propose VITA (VIrtual Transcriptomic Approximation). By aligning H&E and RNA into a joint latent space during training, VITA requires only standard H&E at inference to retrieve morphologically similar historical cases and approximate the continuous RNA signature. Achieving 0.72 classification accuracy and a 0.66 Spearman correlation, VITA provides a cost-effective "virtual transcriptomics" pre-screening tool that preserves the continuous phenotypic spectrum without requiring genomic sequencing.

**Keywords:** Computational Pathology, Cross-Modal Retrieval, Gastric Cancer.

## 1. Introduction

Gastric adenocarcinoma (GAC), a leading global cause of cancer mortality (Sung et al., 2021), exhibits marked histological heterogeneity and poor prognosis. Although immune checkpoint inhibitors (ICIs) have transformed GAC treatment (Janjigian et al., 2021, 2023), variable patient response highlights the failure of current biomarkers (Borcoman et al., 2019) to capture complex tumor immunophenotypes. Identifying the "Inflamed" immunophenotype has been proposed to predict ICI response (Rodriguez et al., 2025), yet its 10-gene RNA signature is costly. Deep learning on standard H&E WSIs offers a cost-effective alternative (Lu et al., 2021); however, current WSI classifiers only offer the binary classification, ignoring the biological phenotypic spectrum and discarding molecular nuance.

To overcome this economic and biological trade-off, we propose VIrtual Transcriptomic Approximation (VITA). VITA leverages expensive paired WSI-RNA data during training to learn an aligned metric space, yet requires only standard H&E at inference to eliminate molecular costs. By framing subtyping as a cross-modal retrieval task, VITA searches this latent space for morphological nearest neighbors. This allows pathologists to bypass genomic sequencing and approximate the underlying continuous RNA signature directly from morphology, yielding an interpretable, zero-cost pre-screening tool.

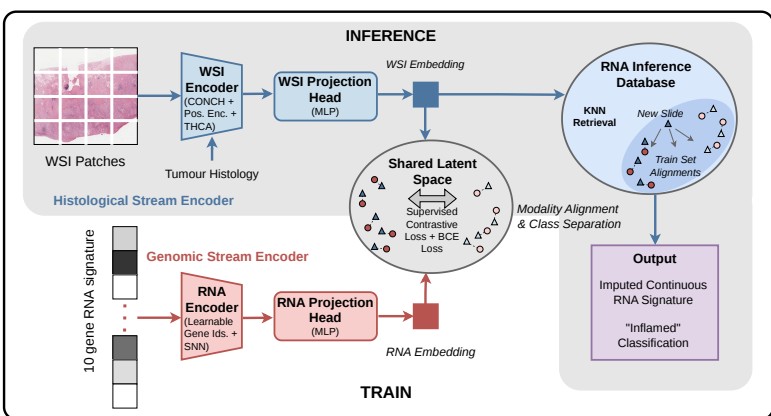

Figure 1: Cross-modal retrieval training and inference framework.

## 2. Methodology

### 2.1. Training and Inference

**Cross-Modal Architecture and Training:** VITA (Fig. 1) employs a dual-stream architecture to learn a continuous joint embedding space. The histological stream extracts visual features via a frozen CONCH backbone (Lu et al., 2024), applying Fourier Positional Encodings and tumour histology-conditioned attention. In parallel, the genomic stream encodes the 10-gene signature using a Self-Normalizing Neural Network (Klambauer et al., 2017). Both modalities are projected into a shared space and optimized jointly using a Supervised Contrastive Loss to transfer genomic knowledge to the visual encoder, alongside a BCE loss to maintain diagnostic separability.

 **Zero-Cost Inference via Cross-Modal Retrieval** At inference, the genomic stream is detached to eliminate molecular costs. Operating unimodally, VITA maps a query H&E slide into the pre-aligned joint space to retrieve its $k = 5$ nearest neighbors, a value determined empirically. Instead of forcing a rigid binary prediction, we aggregate the known RNA profiles of these retrieved cases to impute a continuous RNA signature for the new patient. This search-by-case retrieval framework preserves the spectrum, providing pathologists with an interpretable, zero-cost approximation of the 10-gene RNA signature. Furthermore, while binary classification can be derived directly from this retrieval process, a linear probe is also trained concurrently to perform this specific diagnostic task.

### 2.2. Experimental Setup

We used a curated cohort of $N = 265$ diagnostic WSIs from the TCGA dataset. Ground-truth labels were derived from the continuous 10-gene RNA signature, yielding 142 "Non-Inflamed" and 123 "Inflamed" cases. Additionally, cases were stratified by pathologists according to the histology Lauren classification (Lauren, 1965) for the histological stream. All models were evaluated using 5-fold cross-validation. To validate our retrieval hypothesis, we compared VITA against CLAM (Lu et al., 2021), the leading unimodal WSI classifier and MCAT (Chen et al., 2021), a standard multimodal fusion network for WSI classification.

Table 1: Performance comparison for immunophenotype subtyping.

| Modality | Model | Acc. ↑ | F1 ↑ |
|---|---|---|---|
| Unimodal | CLAM | 0.73 ±0.03 | 0.72 ±0.05 |
| Multimodal | MCAT | 0.61 ±0.07 | 0.66 ±0.06 |
| Cross-Modal | VITA | **0.72** ±0.08 | **0.72** ±0.06 |

Table 2: VITA vs. baseline cross-modal retrieval metrics.

| Model | Query → Target | R@5 ↑ | $\rho_s$ ↑ |
|---|---|---|---|
| CONCH | Test → Train | – | 0.64 ±0.04 |
| (Avg) | Test → Test | 0.42 ±0.05 | 0.62 ±0.08 |
| VITA | Test → Train | – | **0.66** ±0.03 |
| (Ours) | Test → Test | **0.54** ±0.12 | 0.62 ±0.03 |

## 3. Results and Discussion

**Overcoming the Domain Gap:** Direct multimodal fusion (MCAT) struggles in data-scarce regimes (Tab. 1). While unimodal baselines (CLAM) perform well, they rigidly binarize continuous data. VITA addresses this by matching unimodal classification performance while aligning H&E and RNA modalities, setting the foundation for downstream transcriptomic imputation.

**Zero-Cost RNA via Retrieval:** Operating unimodally at inference, VITA suggests the feasibility of zero-cost pre-screening by querying an H&E slide to retrieve morphological neighbors and averaging their ground-truth signatures into an imputed RNA profile. Emulating a real-world clinical

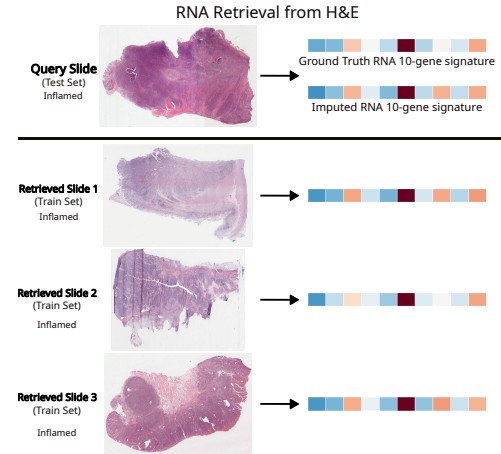

Figure 2: Cross-modal retrieval.

scenario that matches new queries (Test) against a database of known cases (Train), VITA outperforms a baseline of averaged CONCH features (Tab. 2) with superior recall and strong correlation to ground-truth data. Finally, expression barcodes (Fig. 2) validate that VITA effectively reconstructs the phenotypic spectrum without relying on inexact labels.

**Limitations & Future Work:** While VITA matches strong baselines and correlates well with ground-truth RNA, it remains a proof of concept evaluated on one public cohort, and gains are modest. To establish clinical utility, future work will focus on independent validation and exploring advanced aggregation strategies to enhance imputation fidelity.

## 4. Conclusions

We present VITA, a framework predicting immunotherapy response in GAC. By aligning WSI-RNA data during training, it enables unimodal H&E inference via cross-modal retrieval. Imputing continuous RNA signatures from morphological neighbors overcomes multimodal data scarcity and label oversimplification. This interpretable approach shows promising potential to bypass genomic sequencing costs and enable zero-cost pre-screening.

## Acknowledgments

Supported by AGAUR-FI (2025 FI-STEP 00032) from Generalitat of Catalonia/ESF+, PID2023-148614OB-I00 funded by MICIU/AEI/10.13039/501100011033 and EU FEDER.

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
