# OpenReview forum: "Zero-Cost Virtual RNA: Approximating Immunotherapy Signatures via Cross-Modal WSI Retrieval"
_MIDL.io/2026/Short_Papers — MIDL 2026 - Short Papers Poster_

### Official Review · Reviewer_RvjS · 2026-05-03
**Weak Accept: A clinically relevant proof-of-concept, but the novelty is incremental and the retrieval/OOD limitations should be clarified.**

**Rating:** 4
**Confidence:** 4

**Review:**

The experimental evidence is not fully convincing. In Table 1, VITA matches but does not outperform the unimodal CLAM baseline, so the results do not support improved classification performance. Table 2 is intended to support the retrieval/RNA-imputation claim, but the evidence remains limited: R@5 is not clearly defined, Spearman correlation improves only marginally in the Test→Train setting, and there is no Spearman improvement in the Test→Test setting. Since the central claim is virtual RNA approximation, the paper should include clearer and more direct imputation metrics, such as gene-level correlation, signature-score error, or MAE/MSE. The “zero-cost” claim should also be clarified as RNA-free inference for the new patient, since the method still depends on an RNA-annotated reference database. A further concern is out-of-distribution robustness: if a new case has morphology or RNA patterns not represented in the database, VITA will still retrieve nearest neighbors, but those neighbors may not be biologically appropriate. The paper should discuss this limitation and include uncertainty or OOD analysis.

**Summary:**

The paper addresses an important clinical problem: estimating immunotherapy-related RNA signatures from standard H&E slides in gastric adenocarcinoma. VITA is an interpretable cross-modal retrieval framework: it learns from paired WSI-RNA data during training, then uses H&E-only inference to retrieve RNA-annotated neighbors and approximate a continuous RNA signature.

**Strengths:**

Addresses a clinically relevant problem: estimating immunotherapy-related RNA signatures from routine H&E slides.
Proposes an interpretable cross-modal retrieval framework rather than only a black-box binary classifier.
Uses paired WSI-RNA data during training but requires only H&E for the new patient at inference.
Attempts to preserve continuous RNA-signature information instead of reducing everything to a binary label.
Clear short-paper presentation with an easy-to-follow workflow.

**Weaknesses:**

Novelty is incremental; WSI-RNA prediction and multimodal pathology learning have been explored before.
VITA does not outperform the unimodal CLAM baseline in Table 1.
Table 2 provides limited evidence for RNA imputation; Spearman gains are marginal and R@5 is not clearly defined.
The “zero-cost” claim needs clarification because the method still requires an RNA-annotated reference database.
Possible out-of-distribution issue: if a new case has no close match in the database, retrieved neighbors may not be biologically reliable.
Needs more direct RNA-imputation evaluation, such as gene-level correlation, signature-score error, or MAE/MSE.

**Justification Of Rating:**

The paper addresses a clinically relevant problem and presents an interpretable cross-modal retrieval framework for approximating continuous RNA signatures from H&E slides. While the novelty is incremental and the empirical gains are modest, the work is suitable as a short proof-of-concept: it uses paired WSI-RNA data meaningfully during training and enables RNA-free inference for new patients by retrieving from an RNA-annotated reference database.

---

### Decision · Program_Chairs · 2026-05-08

Accept (Poster)